# Effectors of the Type VI Secretion System Have the Potential to Be Modified into Antimicrobial Peptides

Wenjia Lu,[a,b,c,d,e] Hao Lu,[a,b,c,d,e] Chenchen Wang,[a,b,c,d,e] Gaoyan Wang,[a,b,c,d,e] Wenqi Dong,[a,b,c,d,e] ● Chen Tan[a,b,c,d,e]

[a]National Key Laboratory of Agricultural Microbiology, College of Veterinary Medicine, Huazhong Agricultural University, Wuhan, Hubei, China
[b]Hubei Hongshan Laboratory, Wuhan, China
[c]The Cooperative Innovation Center for Sustainable Pig Production, Wuhan, Hubei, China
[d]Hubei Hongshan Laboratory, College of Veterinary Medicine, Huazhong Agricultural University, Wuhan, Hubei, China
[e]Key Laboratory of Preventive Veterinary Medicine in Hubei Province, The Cooperative Innovation Center for Sustainable Pig Production, Wuhan, Hubei, China

**ABSTRACT**  The use of antibiotics has led to the emergence of multidrug-resistant (MDR) bacteria, and there is an urgent need to find alternative treatments to alleviate this pressure. The type VI secretion system (T6SS) is a protein delivery system present in bacterial cells that secretes effectors that participate in bacterial virulence. Given the potential for the transformation of these effectors into antimicrobial peptides (AMPs), we designed T6SS effectors into AMPs that have a membrane-disrupting effect. These effectors kill bacteria by altering the membrane potential and increasing the intracellular reactive oxygen species (ROS) content. Moreover, AMPs also have a significant therapeutic effect both *in vivo* and *in vitro*. This finding suggests that it is possible to modify bacterial components of bacteria themselves to create compounds that fight bacteria.

**IMPORTANCE**  This study first identified and modified the T6SS effector into positively charged alpha-helical peptides. These peptides have good antibacterial and bactericidal effects on G+ bacteria and G− bacteria. This study broadens the source of AMPs and makes T6SS effectors more useful.

**KEYWORDS**  T6SS effector, antimicrobial peptides, multidrug-resistant, Tsap, anti-inflammation

**A**ntibiotic resistance causes difficulties in clinical treatment and results in serious economic losses (1). Antibiotics are vital for treating patients with multidrug-resistant (MDR) bacterial infections (2). From the elucidation of the mechanisms of action of classical first-line drugs to the discovery of the carbapenem resistance gene MCR-1 (3, 4), carbapenems appear to be becoming the last line of defense in the treatment of multidrug-resistant bacteria (5).

To combat the serious challenge posed by MDR, new antibacterial approaches are being sought. These approaches include (i) the synthesis and design of new antibacterial materials (6–10), (ii) the discovery of new uses for old drugs (11–14), (iii) the construction of new drug delivery vehicles (9, 15, 16), and (iv) vaccine development (17–19). AMPs have attracted considerable attention as antimicrobials. The main sources of AMPs are bacteria, archaea, animals (20), fungi, and plants (21). However, the relationship between secretion systems and AMPs remains unclear. In this study, we identified a type VI secretion system (T6SS) effector in extraintestinal pathogenic *Escherichia coli* (*ExPEC*) RS218 and modified it to create an AMP. The results showed that the modified AMPs have a significant inhibitory effect on the growth of bacteria.

T6SS is a protein-delivering nanoweapon that has been found to be present in approximately one-quarter of fully sequenced Gram-negative bacteria (22). Typically, T6SS is involved in the competition between eukaryotes and prokaryotes for survival.

Address correspondence to Chen Tan, tanchen@mail.hzau.edu.cn.

The authors declare no conflict of interest.

When bacteria encounter adversity, they sense the adverse stimuli and rapidly proceed to secrete effectors.

Typically, T6SS secretes effectors that have bactericidal action (23). Some effectors of the T6SS have specific structures; one typical effector structure is a proline-alanine-alanine-arginine (PAAR) structural domain at the N terminus. Effectors with N-terminal PAAR characteristics have variable C-terminal structures (24), and this variability causes the effectors to have different functions. In this paper, we identified an effector in *ExPEC* RS218 with an N-terminal PAAR structure characterized by bacteriocins at its C terminus. Based on this effector, we designed and validated three AMPs that exhibit antimicrobial activity, particularly against Gram-positive bacteria.

Type VI secretion system-related antibacterial peptide (Tsap) AMPs exert a bactericidal effect mainly by disrupting cell membranes and increasing reactive oxygen species (ROS) levels within bacteria. Tsap binds more strongly to lipoteichoic acid (LTA) of Gram-positive bacteria than to lipopolysaccharide (LPS) of Gram-negative bacteria and therefore has a stronger inhibitory effect on Gram-positive bacteria. This study broadens the identified sources of AMPs and describes the modification of AMPs for use as T6SS effectors.

## RESULTS

**Tsap is a T6SS effector in *ExPEC* RS218.** In *ExPEC* RS218, we found a classical T6SS cluster that has a gene with unknown function (gene 1835) located downstream of it (Fig. 1A). The presence of a typical PAAR domain in the 1835 N terminus suggests that this gene might be a potential T6SS effector. We knocked out 1835 and the T6SS essential assembly gene clpV and measured the growth curves (Fig. 1B) of Δ1835, ΔclpV, and the wild-type (WT) strain. There were no differences in the growth curves of the three strains. The results of bacterial competition assays designed to evaluate T6SS function indicated that the survival rate of the prey strain W3110 in the Tsap mutant strain group was significantly increased compared with that of the WT competition group. A similar tendency was observed for the ΔclpV competition group (Fig. 1C). Antiphagocytic activity (Fig. 1D), adhesion, and invasion ability were correspondingly decreased (Fig. 1E and F). These results further suggested that 1835 is a T6SS effector. We recommend that this gene be renamed "type VI secretion system-related antibacterial peptide" (Tsap).

According to the NCBI BLASTP website, the C terminus of 1835 is speculated to be a C39 peptidase. We divided the sequence of 1835 into three parts (Fig. 2A), as follows: a PAAR domain (1 to 324 bp), a C39 peptidase family domain (1041 to 1437 bp), and a middle portion (324 to 1041 bp). Then by detecting cell expression toxicity, according to Fig. 2B, we found that cells show obvious growth inhibition only if the sequences are in the C terminus (Fig. 2B). The C-terminal overexpression strain showed obvious growth inhibition, indicating that this portion of the gene might encode a peptide that is cytotoxic to prokaryotes. We also noted that this portion of the peptide has an $\alpha$-helical structure similar to that found in most AMPs. Based on these findings, we wondered whether this part of the sequence could be modified to produce an AMP.

**Antimicrobial activity.** We designed nine peptides and measured the MICs of those peptides. According to Luna et al. (25), in a nutrient-limited situation, antibacterial peptides confer significantly enhanced antibacterial activity. We measured the antibacterial activity of peptides in RPMI 1640 medium and in RPMI 1640 medium supplemented with amino acids; the MIC results are presented in Table 1. We chose the three peptides that exhibited significant antibacterial activity for further research. These peptides have excellent antibacterial and bactericidal effects against both Gram-negative and Gram-positive bacteria. In comparison to MICs in a single RPMI 1640 medium or other amino acid supplement solution, the inhibitory activity of the bacteria increased with the presence of 0.5 mM L-arginine, 0.5 mM glycine, or nonessential amino acid solutions. Both Tsap1 and Tsap3 significantly inhibited the growth of standard strains, particularly in *Staphylococcus aureus* ATCC 25923. Tsap1 and Tsap3 showed a superior MBC effector, which is 4 to 8 g/mL, compared with Tsap2, which had an MBC of 16 to 32 g/mL. Tsap

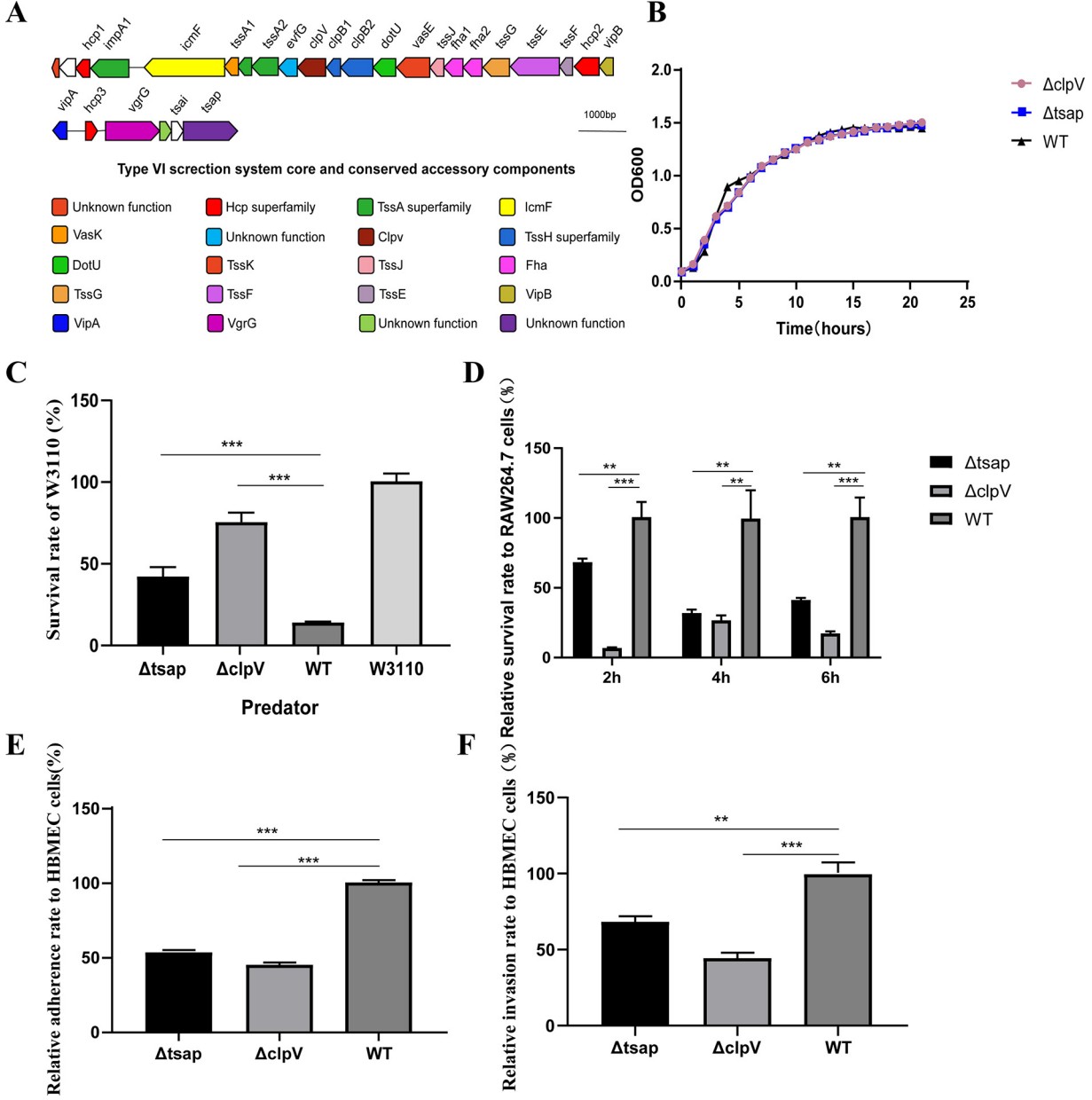

**FIG 1** Tsap is a T6SS effector in *ExPEC* RS218. (A) Schematic diagram of the T6SS gene of RS218. Tsap is located at the end of the T6SS cluster. (B) Growth curves of the Δtsap, ΔclpV, and WT strains. The curves were obtained using a fully automated growth curve analyzer (CFP-1100-C; Finland). (C) Competitive ability of the Δtsap strain. The mutant and WT strains were used as predators, and *E. coli* W3110 was used as prey. The predator and prey were cocultured at a ratio of 1:10 at 30°C for 12 h; 10× serial dilutions were then prepared, and CFUs were counted. (D) Determination of antimacrophage capacity. RAW 264.7 cells were infected with mutant or WT strains at an MOI of 10. After 2 h, 4 h, or 6 h, intracellular bacteria were counted in 10× serial dilutions of the cultures. Adhesion (E) and invasion (F) capabilities of Δtsap are shown. All experiments were repeated three times independently. **, $P < 0.01$; ***, $P < 0.001$.

AMPs also show significant bactericidal activity against MDR strains. The results are presented in Table 2, and the AMPs all have a broad spectrum of antimicrobial activity.

**Antimicrobial peptide design and characteristics.** To provide a better understanding of the nature of the AMPs, we have listed the sequence, number of amino acids, molecular weight, pI, and hydrophobicity (negative values indicate hydrophilicity, and smaller values indicate greater hydrophilicity) in Table 3. In the spiral wheel plots for Tsap-1 (Fig. 3A), Tsap-2 (Fig. 3B), and Tsap-3 (Fig. 3C), different amino acids are shown in different colors. The amino acid linkage sequence is indicated by a progression from black to gray. Through model analysis, we found that the hydrophobic

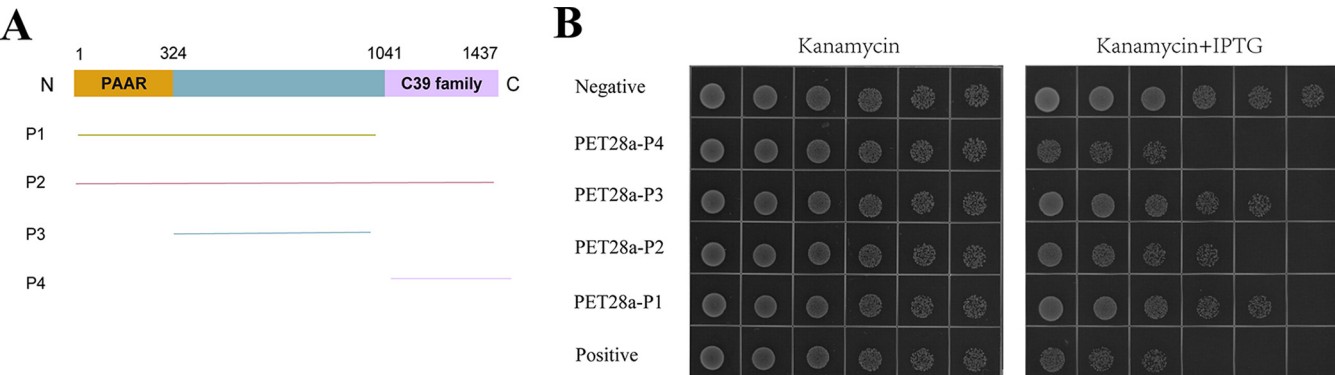

**FIG 2** The C terminus of 1835 has antibacterial activity. (A) Schematic diagram of the 1835 protein. The PAAR region of the peptide is shown in yellow, the middle fragment is shown in blue, and the C39 peptide region is shown in purple. (B) Growth of 10× serial dilutions of bacteria on Luria Bertani agar (LBA) plates and on LBA plates containing 50 μg/mL kanamycin and 0.5 mM isopropyl-β-ᴅ-thiogalactopyranoside (IPTG). The negative control is BL21(DE3) containing an empty PET28a plasmid. The positive control is Rhs-3CT in *ExPEC* PCN033.

amino acids, which are shown in green, lie in the same plane and that the neutral and positively charged amino acids (shown in blue and red, respectively) lie in another plane (Fig. 3D, E, and F). The numbers of different amino acids are shown in Fig. 3G, H, and I.

**Hemolytic activity and cytotoxicity.** In our measurements of the MICs of specific peptides against standard and clinical bacterial strains, we found that Tsap peptides have good efficiency in inhibiting or killing bacteria, especially Gram-positive bacteria. The time-kill curves showed that Tsap-1 can kill *E. coli* and *S. aureus*, especially *S. aureus*, within 7 h. Tsap-2 and Tsap-3 also showed significant bactericidal activity (Fig. 4A and B).

We also measured the cytotoxicity of the peptides to HeLa cells and mouse RAW 264.7 cells, compared with a positive-control melittin-treated group, and the peptides showed only negligible toxicity (Fig. 4C and D). Sheep red blood cells were used to measure the hemolytic activity of those peptides. The results showed that there was only slight hemolytic activity at high peptide concentrations (Fig. 4E). But the group treated with 4 μg/mL melittin already showed complete hemolysis activity.

**Tsap peptides disrupt the membranes of *E. coli* and *S. aureus*.** To determine the mechanism through which the peptides exert their antibacterial activity, we tagged the peptides with fluorescein isothiocyanate (FITC). Using structured illumination microscopy (SIM), which has a higher resolution than confocal or fluorescence microscopy and can visualize the structure of bacteria, we found that the peptides are first located in bacterial membranes and then enter the bacteria (Fig. 5). About all the bacterial membrane was covered by FITC-tagged AMPs after being treated with 0.5 MBC

**TABLE 1** AMP antimicrobial activity against laboratory strains

| Parameter | Antimicrobial activity[a] (μg/mL) | | | | | |
|---|---|---|---|---|---|---|
| | *E. coli* ATCC 25922 | | | *S. aureus* ATCC 25923 | | |
| | Tsap1 | Tsap2 | Tsap3 | Tsap1 | Tsap2 | Tsap3 |
| RPMI | 32 | 128 | 32 | 16 | 64 | 16 |
| Amino acid supplementation in RPMI | | | | | | |
| Nonessential amino acid solution | 16 | 64 | 16 | 16 | 64 | 16 |
| ʟ-Arginine (0.5 mM) | 16 | 64 | 16 | 16 | 16 | 16 |
| Glycine (0.5 mM) | 16 | 64 | 16 | 16 | 64 | 16 |
| Leucine (1 mM) | 32 | 128 | 32 | 16 | 64 | 16 |
| ʟ-Histidine (0.5 mM) | 32 | 128 | 32 | 16 | 64 | 16 |
| ʟ-Tryptophan (0.125 mM) | 32 | 128 | 32 | 16 | 64 | 16 |
| MBC | 4 | 32 | 4 | 4 | 16 | 8 |

[a]All values are MIC unless otherwise specified.

**TABLE 2** AMP antimicrobial activity against MDR bacteria

| Bacterial strain | Antimicrobial activity ($\mu$g/mL) of: | | | | | |
|---|---|---|---|---|---|---|
| | Tsap1 | | Tsap2 | | Tsap3 | |
| | MIC | MBC | MIC | MBC | MIC | MBC |
| *S. aureus* USA300 | 16 | 2 | 16 | 8 | 16 | 2 |
| *S. aureus* 1802043 | 16 | 2 | 16 | 8 | 16 | 2 |
| *S. aureus* USA200 | 64 | 8 | 512 | 64 | 64 | 16 |
| *S. aureus* ATCC 43300 | 16 | 4 | 16 | 8 | 16 | 4 |
| *Streptococcus* SC19 | 16 | 4 | 16 | 8 | 16 | 4 |
| *Bacillus subtilis* NCD-2 | 16 | 4 | 16 | 8 | 16 | 4 |
| *ExPEC* RS218 | 64 | 32 | 256 | 64 | 64 | 32 |
| *ExPEC* PCN033 | 64 | 32 | 256 | 64 | 64 | 32 |

AMPs for 30 min. This finding suggests that Tsap peptides could damage the cell membrane since certain AMPs even reach the membrane of bacteria.

We next used the 2',7'-bis-(2-carboxyethyl)-5-(and-6)-carboxyfluorescein, acetoxymethyl ester (BCECF-AM) probe to measure the hydrogen ion ($H^+$) content of *E. coli* and *S. aureus* cells. Bacteria use respiratory enzymes that are membrane bound to power their energy metabolism. These enzymes absorb chemical energy and convert it across their cell membranes using $H^+$ or $Na^+$ proton pumps. The integrity of the bacterial membrane is thus represented by the $H^+$ content (26). As the concentration of the peptide was increased, the $H^+$ content increased in the Tsap peptide-treated group (Fig. 6A and B), which indicates that Tsap can break the membrane integrity. Several publications claim that various AMPs can cause cell death by raising ROS levels (27) and that ROS levels are also related to the structural integrity of cell membranes (28). The peptides promoted an increase in ROS content (Fig. 6C) and thereby caused bacterial death. These phenomena were especially obvious in *S. aureus*.

The 3,3'-diethyloxacarbocyanine iodide [$DIOC_2(3)$] fluorescent probe produces green fluorescence in bacteria under normal conditions. Higher membrane potentials lead to an aggregation of the dye, causing an increase in red fluorescence. If the membrane potential is altered by disruption of proton channels, less red fluorescence is observed. The flow cytometry results showed that the amount of red fluorescence decreased in *E. coli* ATCC 25922 cells exposed to higher Tsap-1 concentrations. Similar results were obtained in the positive-control carbonyl cyanide *m*-chlorophenylhydrazone (CCCP)-treated group, indicating that Tsap-1 targets the proton motive force in *E. coli*. However, in *S. aureus*, the amount of red fluorescence increased in both the CCCP-treated group and the Tsap-1-treated group, indicating that a change in membrane potential had occurred (Fig. 6D and E).

Next, we performed an ATP leakage assay in which we measured the total and extracellular ATP content of bacteria that had been treated with Tsap peptides for various amounts of time. In *S. aureus* and *E. coli*, the extracellular ATP content increased over time (Fig. 6F and G).

Scanning microscopy was used to visualize membrane integrity in *S. aureus* (Fig. 6H). In the negative-control group, the membranes of *S. aureus* were intact and smooth. However, with exposure to increasing concentrations of peptide for longer times, the bacterial film crumpled or broke.

The experiments described above show that Tsap peptides are able to disrupt the integrity of cell membranes, especially in *S. aureus*.

**TABLE 3** Basic properties of AMPs

| Name | Sequence | No. of amino acids | mol wt | Theoretical pI | Net charge | GRAVY[a] |
|---|---|---|---|---|---|---|
| Tsap1 | WKKLKKMIKK | 10 | 1,330.78 | 10.70 | 6.0 | −1.41 |
| Tsap2 | WKALKKMIMKT | 11 | 1,377.81 | 11.01 | 4.0 | −0.3 |
| Tsap3 | WKALKKMIMKI | 11 | 1,389.87 | 11.01 | 4.0 | 0.17 |

[a]GRAVY indicates grand average of hydropathy.

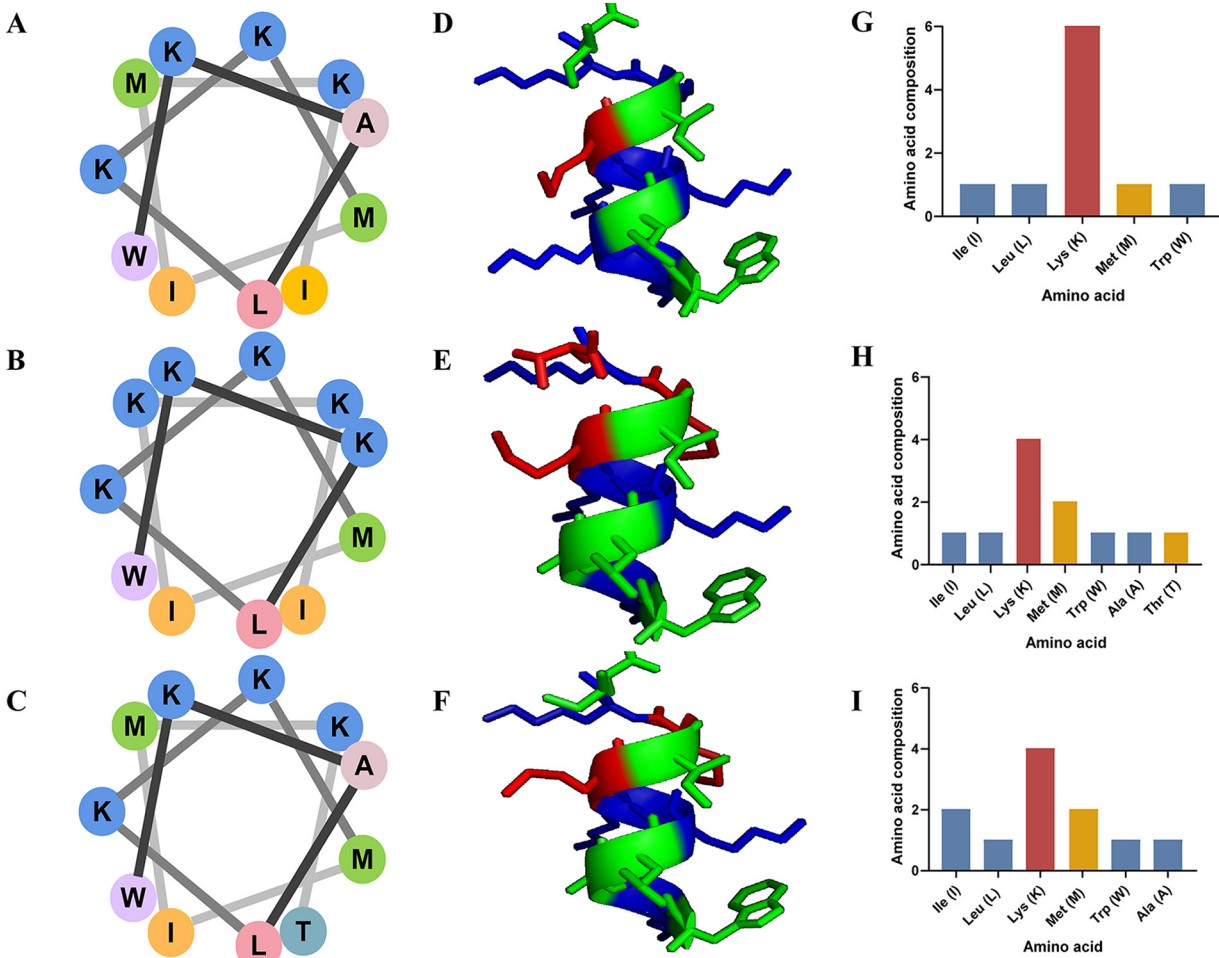

**FIG 3** Antimicrobial peptide and characteristics. Spiral wheels for Tsap-1 (A), Tsap-2 (B), and Tsap-3 (C). Different colors indicate different types of amino acids; hydrophobic amino acids are shown in peach, yellow, green, pink, and purple; alkaline amino acids are shown in blue; and neutral (uncharged) amino acids are shown in bluish green. The 3D model of Tsap-1 (D), Tsap-2 (E), and Tsap-3 (F). The amino acid assembly of Tsap-1 (G), Tsap-2 (H), and Tsap-3 (I) is shown. Alkaline amino acids were shown in red, hydrophobic uncharged residues were colored in yellow, and polar uncharged amino acids were marked in blue.

**Tsap peptides bind preferentially to _S. aureus_ lipoteichoic acid (LTA).** LTA is a membrane component of Gram-positive bacteria, and its role is comparable to that of LPS in Gram-negative bacteria. We used isothermal titration calorimetry (ITC) to detect the bonding energy of peptides to the _E. coli_ and _S. aureus_ membrane components LPS (Fig. 7A to C) and LTA (Fig. 7D to F). The binding energy of the Tsap peptides to LTA was much higher than that of their binding to LPS at the same concentrations.

We then performed a competitive bactericidal assay in which we added 1 mg/mL LPS or 1 mg/mL LTA to cultures of ATCC 25923. As the concentration of Tsap peptides was increased, the survival of the LPS-neutralized groups obviously decreased (Fig. 7G to I). However, there was no obvious bactericidal effect in the LTA-neutralized groups. This result means that LTA neutralized most of the peptides and indicates that Tsap peptides have a higher affinity for LTA than for LPS.

**Tsap peptides suppress LTA-stimulated inflammation _in vitro_ and _in vivo_.** Both LPS and LTA induce inflammation similar to the inflammation that occurs during the bacterial infection process (29). Our results showed that the endotoxin was neutralized as the Tsap concentration increased, and at concentrations approaching 100 $\mu$g/mL, endotoxin was almost completely neutralized (Fig. 8A to C) (30, 31). After treatment of HeLa cells with 200 ng/mL LTA for 8 h, the levels of tumor necrosis factor alpha (TNF-$\alpha$) (Fig. 8D to F) and interleukin-6 (IL-6) (Fig. 8G-I) in the Tsap peptide-treated group were much lower than those in the untreated group. This effect was enhanced as the

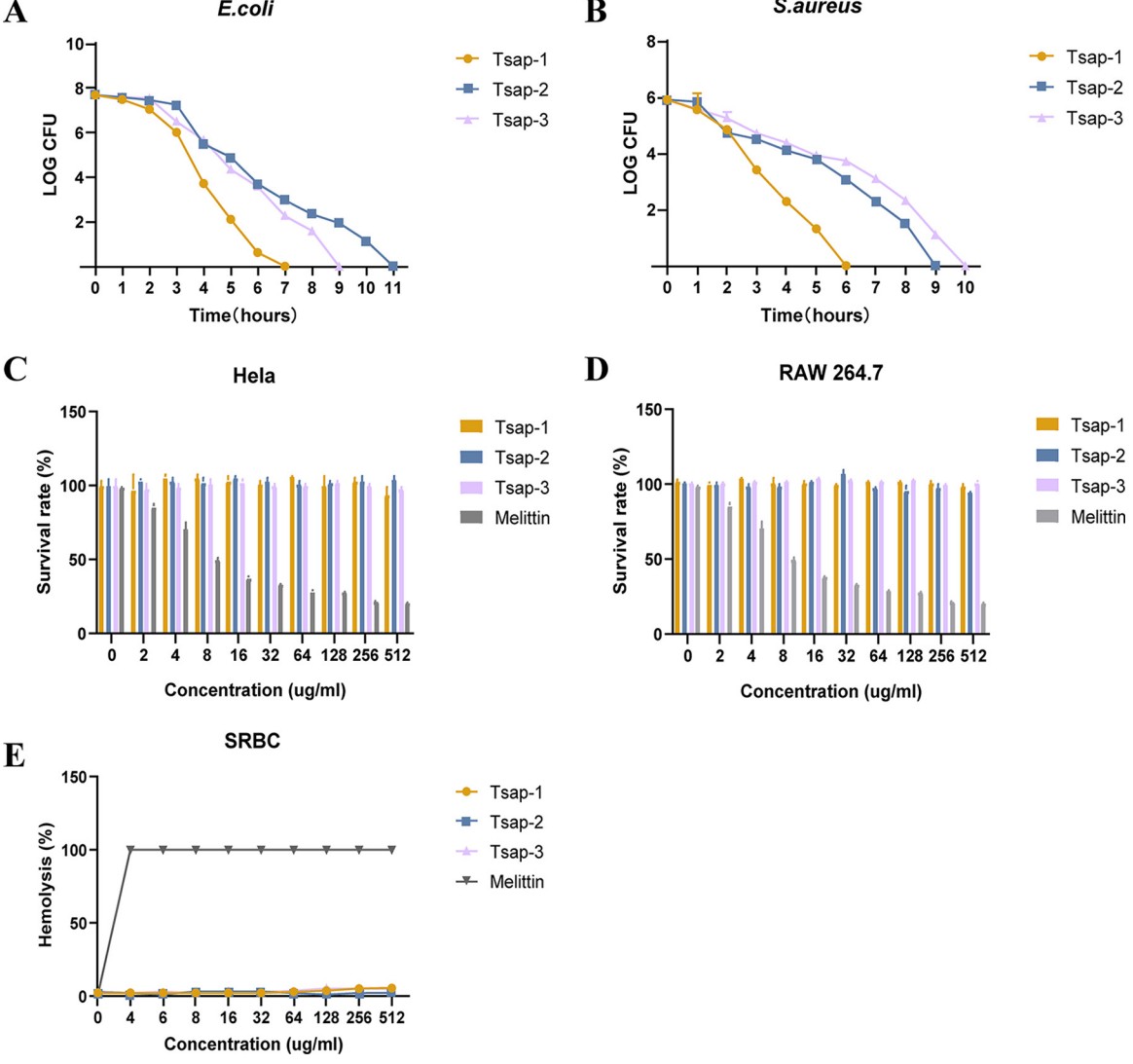

**FIG 4** Hemolytic activity and cytotoxicity. Shown are time-kill curves for Tsap AMPs (1× MBC) in *E. coli* ATCC 25922 (A) and *S. aureus* ATCC 25923 (B). Cytotoxicity of AMPs at various concentrations in HeLa cells (C) and RAW 264.7 cells (D) is shown. Notably, 0 $\mu$g/mL to 512 $\mu$g/mL of melittin were used as a positive control, and the PBS group was used as a negative control. Hemolytic activity toward sheep red blood cells of Tsap peptides at various concentrations are shown (E), and melittin was used as the control.

concentration of Tsap was increased. After measuring the levels of TNF-$\alpha$ and IL-6 *in vitro*, we measured the effects of peptides *in vivo*.

We used the LTA-induced mouse endotoxemia model to detect inflammation in lung tissue (Fig. 8J to L). LTA pulmonary inflammatory mice displayed significant alveolar epithelial and capillary endothelial cell injury, alveolar edema, and inflammatory cell infiltration. The treated group showed a significant reduction in those changes. The results indicate that Tsap AMPs can alleviate LTA-stimulated inflammation *in vitro* and *in vivo*.

**Tsap-1 has a good combined bactericidal effect *in vitro* and *in vivo*.** Ciprofloxacin is a broad-spectrum antibacterial drug that is active against Gram-negative and Gram-positive bacteria and is used widely in the treatment of bacterial infections (31). We found previously that the MICs of ciprofloxacin, Tsap-1, Tsap-2, and Tsap-3 in *S. aureus* USA200 were 16 $\mu$g/mL, 64 $\mu$g/mL, 512 $\mu$g/mL, and 64 $\mu$g/mL, respectively. The effects of the administration of Tsap peptides in combination with ciprofloxacin are shown as a checkerboard plot. The fractional inhibitory concentration indices (FICIs) measured in the assays were 0.5, 1.125, and 1.25. Only Tsap-1 showed a synergistic

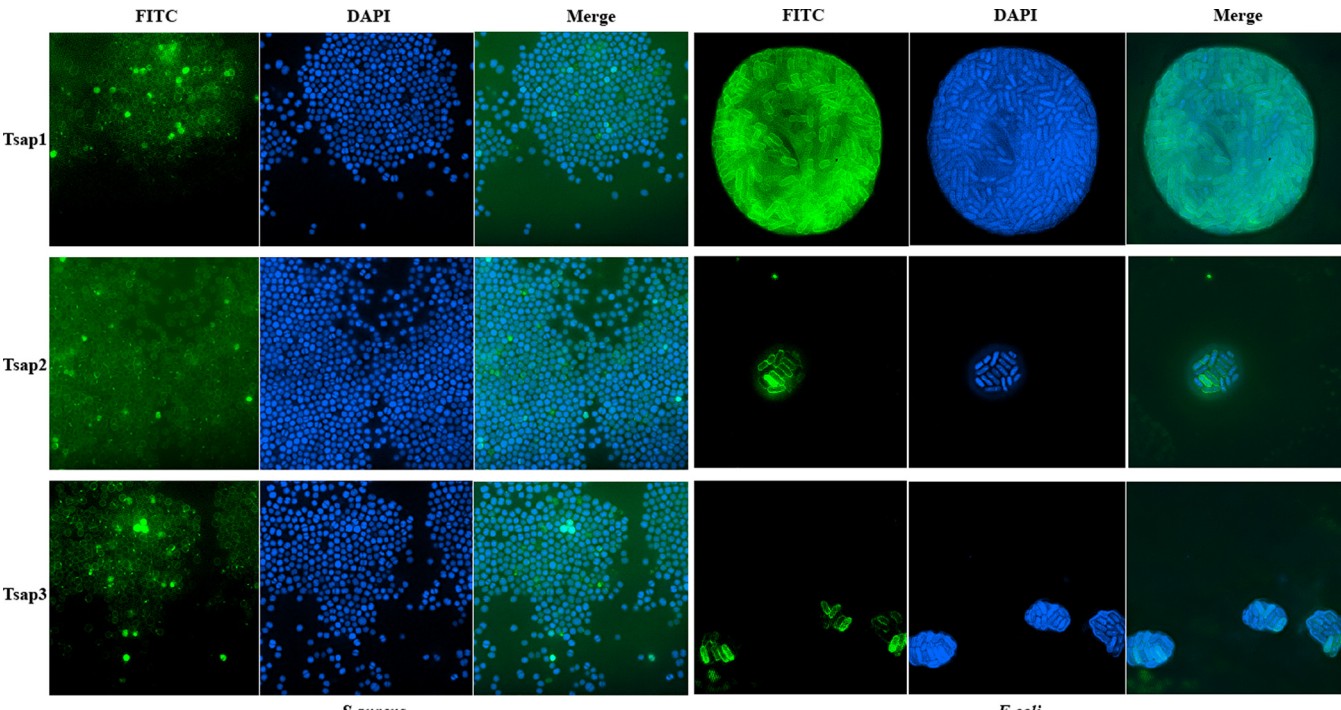

**FIG 5** Colocalization of FITC-labeled Tsap AMPs with *E. coli* and *S. aureus*. Bacteria at mid-log phase were incubated with 0.5× MBC FITC-labeled AMPs for 30 min at 37°C. The bacteria were then incubated with DAPI for 15 min and observed using SIM, and images were acquired.

effect with ciprofloxacin; no differences were observed when Tsap-2 or Tsap-3 was combined with ciprofloxacin.

In the *S. aureus*-infected mouse model, all of the mice in the untreated group died within 3 days. Three mice in the Tsap-1-treated group and four mice in the ciprofloxacin-treated group died within 7 days, indicating 70% and 60% protection, respectively, by the drug. However, in the group that was treated with a combination of Tsap-1 and ciprofloxacin, 90% protection was achieved (Fig. 9B).

After infection with *S. aureus* USA200, mouse tissue CFU loads were counted. Although the CFU load of the Tsap-1-treated group did not differ from that of the ciprofloxacin treatment group, the bacterial loads in all tissues of the animals in the treated groups reduced by about 32 times compared with those of the untreated group, and those of the animals treated with a combination of the two drugs decreased by 1,000-fold. The experiments described above show that Tsap-1 has good *in vivo* and *in vitro* bactericidal effects and that it has a strong antibacterial effect when administered in combination with ciprofloxacin.

## DISCUSSION

AMPs are naturally abundant peptides that can be obtained from a wide range of sources and vary in length from a few to several tens of amino acid residues. T6SS is a nanoweapon that is present in many Gram-negative bacteria, and one of its functions is to secrete effectors that have bactericidal properties. Normally, bacteria evolve corresponding proteins to protect themselves. Although the direct expression of such proteins is of little significance, it indicates that they have the potential to be modified into AMPs. In this study, we identified a T6SS effector (Tsap) in *ExPEC* RS218, modified it after sequence analysis, and investigated the antimicrobial mechanisms of action of the three resulting AMPs (Tsap-1, Tsap-2, and Tsap-3).

In addition to their structural differences, AMPs also differ in their physicochemical properties (32). The $\alpha$-helix is one of the most typical structures in AMPs, and the original amino acid sequence of Tsap "WDALKKMIMDT" was modified into three AMPs. Sequence analysis and the construction of models of the three AMPs showed that all

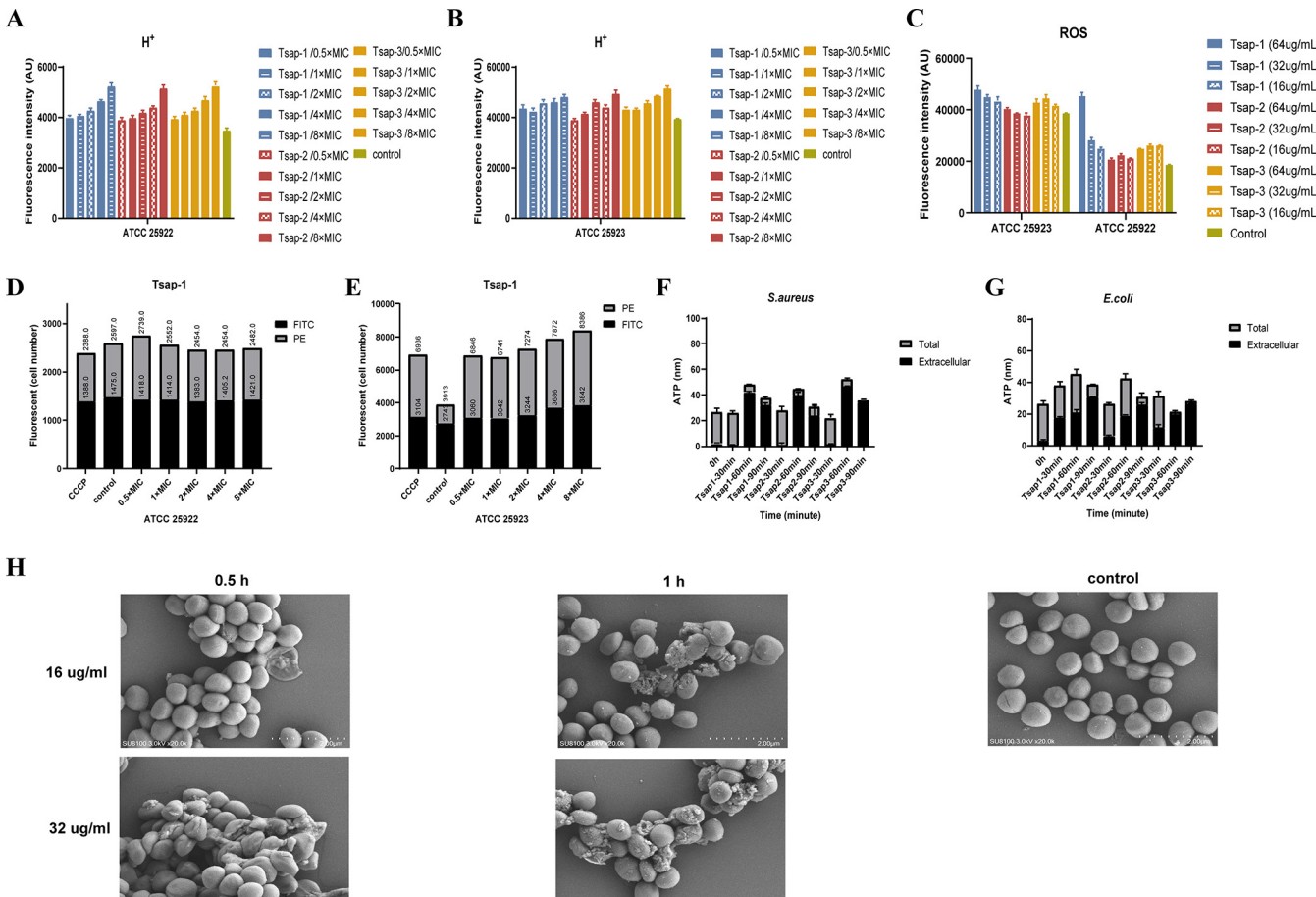

**FIG 6** Tsap peptides disrupt the membranes of *E. coli* and *S. aureus*. The H+ content of ATCC 25922 (A) and ATCC 25923 (B) after treatment with Tsap peptides at various concentrations is shown. Untreated bacteria were used as a control. (C) ROS levels in ATCC 25922 and ATCC 25923 after treatment with Tsap AMPs at various concentrations. The effect of Tsap on membrane permeability in *Escherichia coli* (D) and *Staphylococcus aureus* (E) is shown. FITC staining and flow cytometry were used to determine the number of bacteria with altered membrane potential (PE) and the number of unaltered bacteria after exposure to Tsap-1 at 0.5× MIC, 1× MIC, 2× MIC, 4× MIC, and 8× MIC. The CCCP-treated group was used as a positive control. Determination of ATP leakage in *S. aureus* (F) and *E. coli* (G) treated with AMPs for different amounts of time is shown. The bacterial cells were treated with Tsap AMPs at 1× MIC for 30 min, 60 min, and 90 min, and intracellular and extracellular ATP contents were then determined. (H) Shows scanning microscopy images of Tsap-1-treated *S. aureus*. Scanning microscopy images of *S. aureus* ATCC 25923 are also shown after treatment with Tsap-1 at 16 µg/mL or 32 µg/mL for 0.5 h or 1 h, respectively. Untreated ATCC 25923 were used as a negative control.

three AMPs have $\alpha$-helical structures and consist mainly of hydrophobic and neutral amino acid residues (33). Both Tsap-1 and Tsap-2 are hydrophilic, and Tsap-1 is especially hydrophilic. In general, cationic AMPs have better antimicrobial activity than non-cationic AMPs. All three of the modified AMPs are positively charged, which is one of the reasons for the good antimicrobial efficacy of these three antimicrobial peptides.

Furthermore, we investigated the antibacterial mechanism of action of the Tsap AMPs. We found that all three AMPs have good antibacterial and bactericidal activity against both Gram-positive bacteria and Gram-negative bacteria. The effect on Gram-positive bacteria was particularly pronounced. It was manifested mainly by membrane damage and an elevation of ROS levels. The AMPs had a stronger scavenging effect on Gram-positive bacteria due to their higher affinity for LTA.

The LPS of Gram-negative bacteria plays an important role in the inflammatory process, and the LTA component of Gram-positive bacteria has a role similar to that of LPS (34). We first assessed the toxicity of Tsap AMPs in cytotoxicity assays. The three peptides were found to have almost no toxic effects on cells. Second, through the establishment of an *in vitro* LTA cell inflammation model, we found that Tsap alleviates the inflammation caused by LTA. In addition, the three peptides alleviated lung inflammation caused by the administration of LTA to mice.

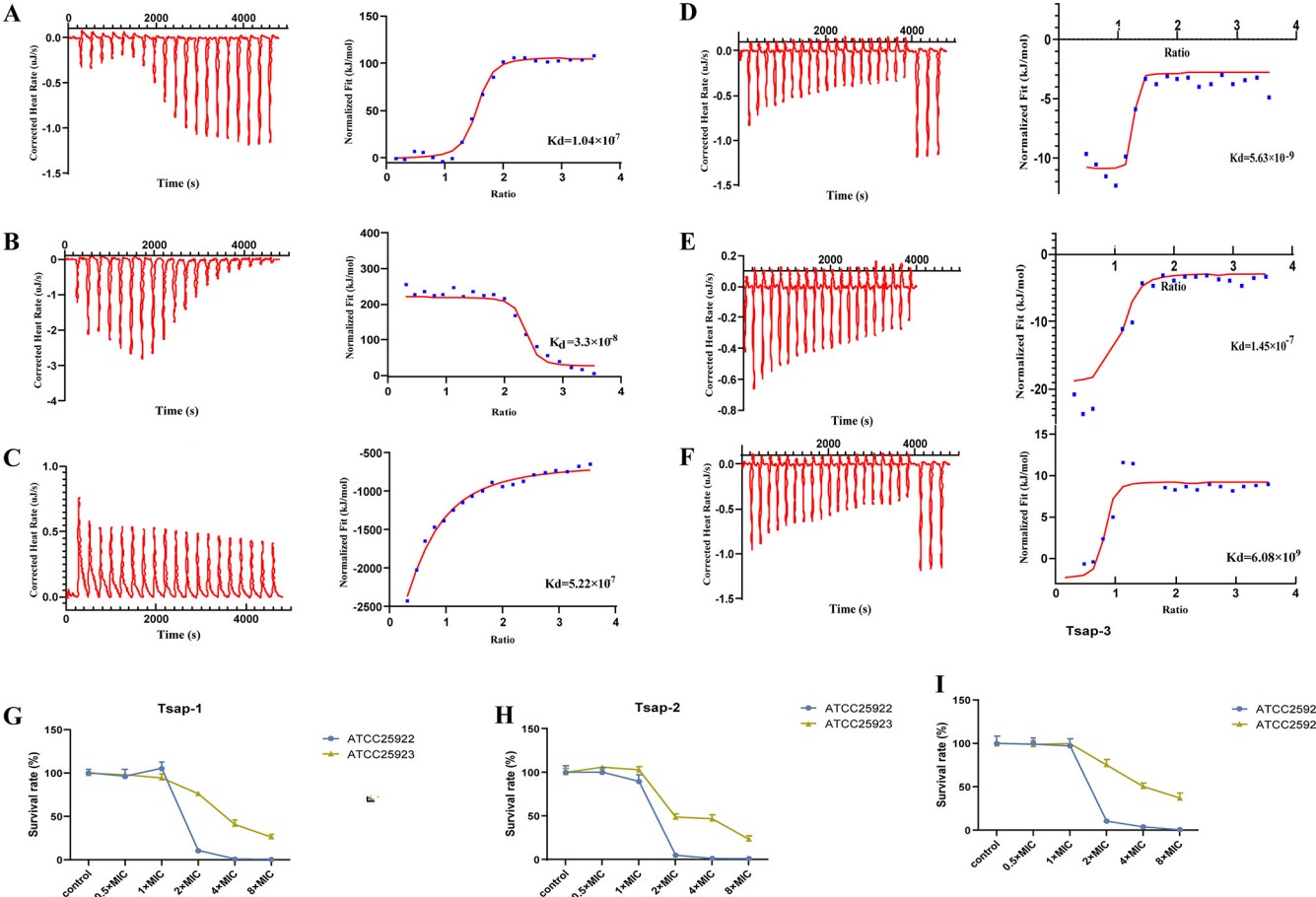

**FIG 7** Tsap peptides exhibit preferential binding to *S. aureus* LTA. The ITC results of LPS binding to Tsap-1 (A), Tsap-2 (B) and Tsap-3 (C). ITC results for Tsap-1 (D), Tsap-2 (E), and Tsap-3 (F) and LTA are shown. The original titration data and the integrated heat measurements are shown in the left and right plots, respectively. Competitive bactericidal activity of Tsap-1 (G), Tsap-2 (H), and Tsap-3 (I) between LPS and LTA. *S. aureus* ATCC 25923 cells were grown to mid-log phase, washed three times with PBS and adjusted to an $OD_{600}$ of 1.0. LPS (1 mg/mL) and LTA (1 mg/mL) were added separately, and various concentrations of AMPs were added. After incubation at 37°C for 2 h, 10× serial dilutions were prepared, and CFU were counted.

The kidneys' ability to metabolize substances can be diminished by bacterial infections and chronic diseases; in individual with these conditions, antibiotic doses must be reduced, and bacterial killing is achieved by other means (35). To assess whether the Tsap AMPs were effective in combination with first-line drugs, we used ciprofloxacin in combination with the AMPs against the MDR strain *S. aureus* USA200. Only Tsap-1 displayed synergy with ciprofloxacin. The alleviating effect of the AMPs on lung infections caused by LTA suggests that these three peptides may have *in vivo* antimicrobial activity. We infected mice with *S. aureus* USA200 and found a significant decrease in the tissue load of *S. aureus* in mice in the Tsap-1 and ciprofloxacin treatment groups, but there was no significant difference between these two groups. This finding suggests that both AMP and ciprofloxacin have a good *in vivo* clearance effect. The animals that received a combined treatment with Tsap-1 and ciprofloxacin showed better bacterial clearance, indicating that the AMP and the antibiotic had a synergistic antibacterial effect.

In summary, this study identified and modified the T6SS effector into the following three positively charged alpha-helical peptides: Tsap-1, Tsap-2, and Tsap-3. These peptides have good antibacterial and bactericidal effects on Gram-positive bacteria and Gram-negative bacteria. The first step in the antibacterial mechanism of Tsap AMP is molting on the cell membrane and interaction with LPS or LTA. This process leads to ion efflux and bacterial death through alterations in membrane potential and ion permeability. Bacterial death is then promoted by the elevation of intracellular ROS levels.

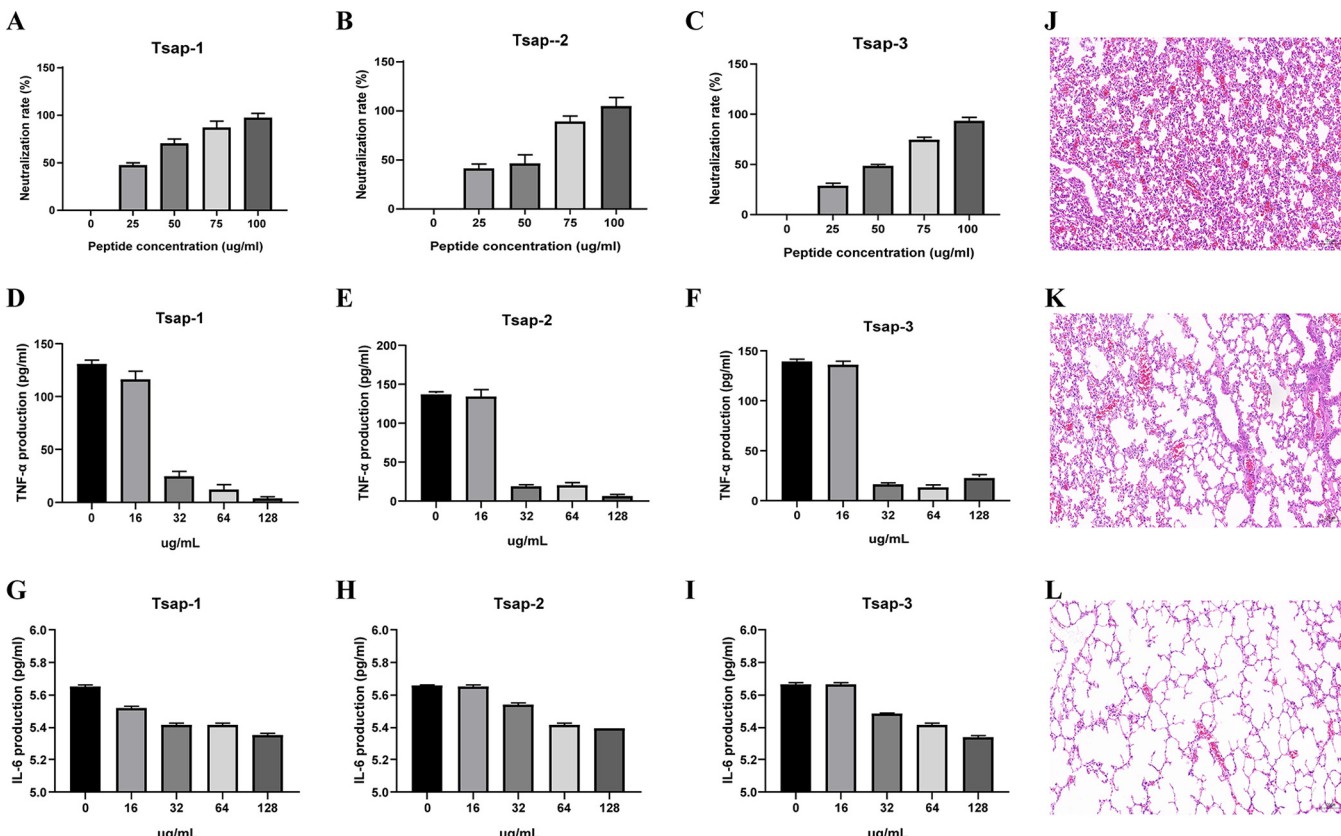

**FIG 8** Tsap peptides suppress LTA-stimulated inflammation *in vitro* and *in vivo*. The ability of Tsap-1 (A), Tsap-2 (B), and Tsap-3 (C) to neutralize endotoxins at various concentrations is illustrated. TNF-$\alpha$ content in the Tsap-1-treated (D), Tsap-2-treated (E), and Tsap-3-treated (F), LTA-infected inflammation model. IL-6 content in Tsap-1-treated (G), Tsap-2-treated (H), and Tsap-3-treated (I), LTA-infected HeLa cell inflammation models. LTA-mediated lung inflammation in mice (J) and Tsap-1-treated sections (K); the untreated group was used as a control (L).

During this period, LPS and LTA released by bacteria can be neutralized by Tsap, yielding an anti-inflammatory effect. The synergistic effect of this AMP with antibiotics also suggests that Tsap can function as an antimicrobial additive.

## MATERIALS AND METHODS

**Bacteria and materials.** The strains used in this work are listed in Table S1 in the supplemental material. *E. coli*, *Bacillus subtilis* NCD-2, and *S. aureus* were cultured in LB broth, and *Streptococcus* was cultured in Trypticase soy broth (TSB) supplemented with 10% fetal bovine serum (FBS) at 37°C. RPMI 1640 medium was purchased from Gibco (USA). Chloromycin, isopropyl-$\beta$-D-thiogalactoside, 2′,7′-dichlorofluorescein diacetate (H$_2$DCFDA), ciprofloxacin, LPS, and LTA were purchased from Merck (USA).

**Gene manipulation.** CRISPR-Cas9-mediated mutant construction was performed as described previously (36). *ExPEC* RS218 was transformed with pCas, and an LBA plate containing 50 $\mu$g/mL kanamycin was used to select the positive strain. Electrocompetent receptor cells were prepared in 10 mM L-arabinose. We used the RS218 genome as a template and primer up-1 (GGGGTTGGGCCAGACGGTGAATGTG) and primer up-2 (ACCATCCCCGATATGATAGTTGCTTATTGATTCCTGAATA) to obtain the upstream homologous arm and primer down-1 (TATTCAGGAATCAATAAGCAACTATCATATCGGGGATGGT) and primer down-2 (CATTTTTAAC CTCAGGTGAAATA) to obtain the downstream homologous arm. Fusion PCR was used to connect the upstream and downstream homologous arms. We then used the pTarget plasmid as a template, and primer-1 (GGTAGTGCTCATGCCCCTTCGTTTTAGAGCTAGAAATAGC) and primer-2 (GAAGGGGCATGAGCACTACCAC TAGTATTATACCTAGGAC) were used to obtain the recombinant plasmid. The fusion homologous arm and the recombinant plasmid were then transformed into the pCas-positive strain. After culture at 30°C overnight in 50 $\mu$g/mL kanamycin and 50 $\mu$g/mL spectinomycin, the recombinant strain was identified by PCR.

**Bacterial competition assay.** Bacterial competition assays were performed as described previously (37). The mutant and WT strains were used as predators, and *E. coli* W3110 was used as prey. Plasmid PHSG396 containing a chloramphenicol resistance gene was transferred into W3110. The bacteria were then incubated until the culture reached an optical density at 600 nm (OD$_{600}$) of 0.9, washed three times with phosphate-buffered saline (PBS), adjusted to an OD$_{600}$ of 0.5, and mixed at a predator:prey ratio of 1:10. The mixture was spotted onto a 50% LBA nitrocellulose membrane (Millipore). After incubation on plates at 30°C for 12 h, the bacteria were washed and counted after 10-fold serial dilution. The experiment was repeated at least three times.

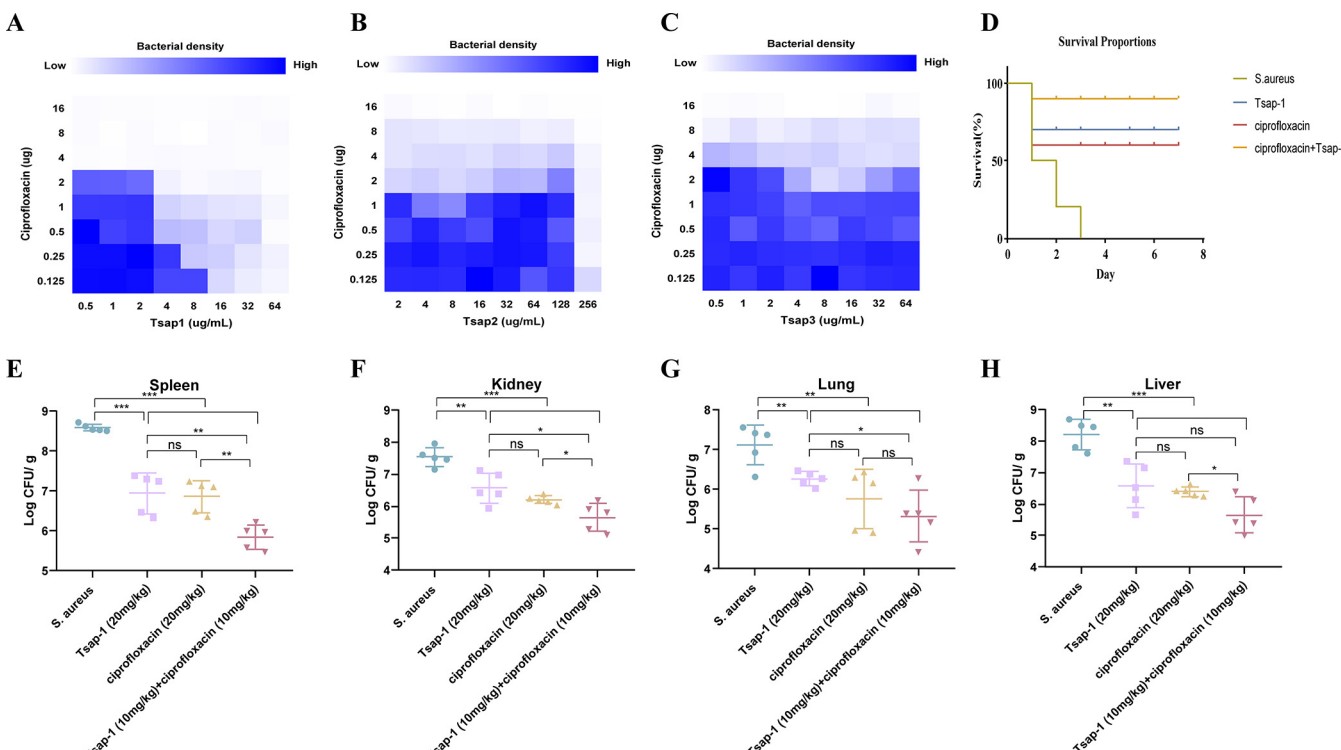

**FIG 9** Tsap-1 exhibits good combined bactericidal effects *in vitro* and *in vivo*. Schematic checkerboard diagram showing the results obtained when Tsap-1 (A), Tsap-2 (B), and Tsap-3 (C) were administered in combination with ciprofloxacin. The blue color ranging from light to dark indicates an increase in drug concentration. Survival of mice after *S. aureus* USA200 injection in the ciprofloxacin, Tsap-1, and combination treatment groups (D). Bacterial load in spleen (E), kidney (F), lung (G), and liver (H) in mice infected with *S. aureus* USA200 after treatment with ciprofloxacin, Tsap-1, or a combination of the two.

**Measurement of adhesion, invasion, and antiphagocytic activity.** Adhesion and invasion were assessed as described previously (38). Human brain microvascular endothelial cells (HBMECs) were inoculated into 24-well cell culture plates. After the cells had covered the bottom of the plate, they were cultured in RPMI 1640 medium without FBS for 30 min. The bacterial concentration was then adjusted, and the cells and bacteria were cocultured at a multiplicity of infection (MOI) of 10. For the invasion assay, the plates containing the cells were centrifuged at 1,000 × *g* for 10 min (this step was omitted in the adhesion assay). The mixture was then incubated for 1 h. For the invasion assay, 100 $\mu$g/mL gentamicin was added, and the cells were incubated for 1 h. The plates were then washed three times with PBS and reincubated in 1640 medium supplemented with 5% FBS for 1 h. After three washes with PBS, the cells were lysed with 0.025% Triton X-100, and 10-fold dilutions were plated onto smear plates and counted. The experiment was repeated at least three times.

Antiphagocytic activity was assessed as described previously with some modifications (39). In brief, mouse macrophage RAW264.7 cells were cultured and plated onto 24-well plates. When the cells had covered the bottom of the plate, the cultures were incubated in Dulbecco's modified Eagle's medium (DMEM) without FBS for 30 min. The bacterial concentration was then adjusted, and the cells were cocultured with bacteria at an MOI of 10. The plates containing the cells were then centrifuged at 1,000 × *g* for 10 min. After incubation for 1 h, the cells were washed three times with PBS and incubated in DMEM containing 100 $\mu$g/mL gentamicin for 1 h, followed by three washes with PBS. The cells were then incubated for 2, 4, or 6 h; washed three times with PBS; lysed in 0.025% Triton X-100; and diluted and spread onto plates for counting. Survival was calculated using the following equation: [(CFU/mL) $t$ = 2/4/6 h/(CFU/mL) $t$ = 0 h] × 100%.

**Peptides.** Tsap-1 (WKKLKKMIKKWKKLKKMIKK), Tsap-2 (WKALKKMIMKT), Tsap-3 (WKALKKMIMKIWKAL KKMIMKI), and melittin (GIGAVLKVLTTGLPALISWIKRKRQQ) were synthesized by the GenScript Biotechnology Company (Nanjing, China).

**MIC and MBC.** MIC was measured according to Luna et al. (25) as described previously. Briefly, 100 $\mu$L of RPMI 1640 medium containing drugs at various concentrations was cultured with 100 $\mu$L of bacteria at 1 × 10⁶ CFU/mL in 96-well plates at 37°C for 18 to 20 h. Wells containing 100 $\mu$L of bacteria at 1 × 10⁶ CFU/mL and 100 $\mu$L medium were used as positive controls. Wells containing 100 $\mu$L drug and 100 $\mu$L medium were used as drug-negative controls, and wells containing 200 $\mu$L medium were used as negative controls.

In amino acid-supplemented RPMI 1640 medium, 100× nonessential amino acid solution, L-arginine (0.5 mM), glycine (0.5 mM), leucine (1 mM), L-histidine (0.5 mM), and L-tryptophan (0.125 mM) were added separately.

The MBC, in which a combination of 10 $\mu$L from MIC was put into Mueller-Hinton agar medium and incubated for 24 h, was the first one in which bacterial growth was seen.

**Time-kill curve assay.** The time-kill curve assay was performed as described previously (40). Briefly, bacteria that had been cultured overnight were transferred to LB broth, grown to an $OD_{600}$ of 0.5, washed three times with PBS, and adjusted to $1 \times 10^6$ cells/mL. Peptide ($1\times$ MBC) was added to the bacterial culture, and 100-$\mu$L aliquots of the culture were removed hourly thereafter. The CFUs were counted after $10\times$ serial dilution.

**Cytotoxicity and hemolytic activity.** The cytotoxicity of the peptides to HeLa and RAW 264.7 cells was measured using CCK8. Cells were seeded in 96-well plates at $10^4$ cells/well, and various concentrations of Tsap peptides were added to the fully grown cells. Cells that had been exposed to melittin at various concentrations were used as positive controls, and wells containing PBS only were used as negative controls. After incubation for 2 h, the cells were washed three times with PBS, and the medium was replaced with fresh medium containing 10% CCK8 solution. After incubation of the cells for an additional 3 h, the $OD_{590}$ of the culture was measured using a microplate reader with a multiwavelength measurement system (Fluostar Omeg, USA). Each measurement was taken in triplicate.

Sheep red blood cells (SRBCs) were used to detect hemolytic activity according to a previous method. Briefly, fresh SRBCs were washed with PBS, and a suspension of 10% SRBCs was mixed with solutions of Tsap or melittin at various concentrations. After incubation at 37°C for 6 h, the cells were removed by centrifugation at $800 \times g$ for 10 min, and the absorbance of the supernatant at 540 nm was measured. Culture supernatant from cells that had been stimulated with 1% Triton X-100 was used as a positive control, and culture supernatant from cells that had been incubated with PBS was used as a negative control. Each measurement was taken in triplicate.

**Structured illumination microscopy (SIM) assay.** The location of Tsap AMPs in bacteria was visualized by SIM. Bacteria were cultured overnight, grown to log phase, washed three times with PBS, and coincubated with FITC-labeled AMP at a concentration of $0.5\times$ MBC for 30 min. After being washed three times with PBS, the bacteria were stained with 4′,6-diamidino-2-phenylindole (DAPI; Sigma) for 30 min at 37°C. The cells were then washed three times with PBS, and colocalization of Tsap and bacteria was observed by SIM (Nihonika, Japan).

**Determination of ROS content.** *E. coli* and *S. aureus* were incubated overnight, transferred, grown to an $OD_{600}$ of approximately 0.5, washed three times with PBS, adjusted to an $OD_{600}$ of 1.0, incubated with various concentrations of AMPs and bacteria at 37°C for 1 h, and washed three times with PBS. The cell precipitate was collected and resuspended in PBS containing $H_2DCFDA$. A total of 200 $\mu$L of cell suspension was added to black 96-well plates, and a microplate reader with a multiwavelength measurement system (PE, USA) was used at an excitation wavelength of 488 nm and an emission wavelength ($\lambda$) of 507 nm. Each measurement was taken in triplicate.

**Membrane potential measurements.** Changes in membrane potential after exposure to peptides were measured using $DIOC_2(3)$ as described previously with slight modifications (41). Specifically, after an overnight incubation of the bacteria, the bacteria were transferred to fresh medium and allowed to grow to mid-log phase. They were then collected by centrifugation at 5,000 rpm for 10 min, washed three times with PBS, adjusted to an $OD_{600}$ of approximately 1.0, and incubated with various concentrations of AMPs or 5 $\mu$m CCCP for 1 h. After three washes with PBS, the cells were added to PBS containing $DIOC_2(3)$ and incubated for 30 min at 37°C. After three more washes with PBS, the numbers of red (PE) and green fluorescent (FITC) bacteria were analyzed by flow cytometry (BD, USA). More than 10,000 bacteria were analyzed in each experiment. The results are expressed as the numbers of green and red fluorescent bacteria obtained at different concentrations of AMPs. The CCCP-treated group was used as the positive control.

**Measurement of ATP content.** Total and extracellular ATP content was measured using an ATP assay kit (Beyotime, China). The protocol was performed according to the manufacturer's instructions (42). Briefly, bacteria were incubated overnight, transferred to fresh medium, and allowed to grow to mid-logarithmic phase. They were then washed three times with PBS and adjusted to an $OD_{600}$ of 1.0. After exposure to various concentrations of AMPs at 37°C for 1 h, the supernatants and the precipitates of the cultures were collected separately. The precipitates were digested to produce a cell lysate, which was then added to an ATP detection reagent. A total of 200 $\mu$L of cell suspension was added to black 96-well plates, and a microplate reader with a multiwavelength measurement system (PE, USA) was used at an excitation wavelength of 488 nm and an emission wavelength ($\lambda$) of 507 nm. Each measurement was taken in triplicate.

**SEM assay.** Bacteria were cultured overnight at 37°C in LB medium and grown to logarithmic cell phase after being recultured. The bacteria were washed three times with PBS and treated with Tsap-1 at 16 $\mu$g/mL and 32 $\mu$g/mL for 30 min or 1 h. The culture was then centrifuged at $4,000 \times g$ for 10 min. After three washes with PBS, the cells were fixed for 2 h at room temperature in electron microscope fixative and then fixed overnight at 4°C. The cells were then dehydrated, sprayed with gold, and visualized by scanning electron microscopy (Regulus 8100; Hitachi, Japan).

**Isothermal titration calorimetry (ITC).** The binding of Tsap AMP to LPS/LTA was measured by ITC at 25°C using a Nano ITC instrument. AMP (0.1 mM/mL), LPS (1 mM), and LTA (1 mM) were dissolved in PBS. A total of 300 $\mu$L of the AMP solution was added to the bacterial culture together with either LPS or LTA (volume, 50 $\mu$L). The samples were aspirated and titrated 20 times (2.5 $\mu$L each time), and data were collected. The Tsap solution was titrated with PBS to correct for the heat of dilution. The data were analyzed using NanoAnalyze software, and binding was determined using an independent binding model. All ITC experiments were performed in triplicate.

**LPS/LTA competitive bactericidal assay.** The LPS/LTA competitive bactericidal activity of the peptides was evaluated using the experimental protocol described previously (28). *S. aureus* ATCCw25923

was used as a target for competitive bactericidal activity. Peptides were added to bacterial cultures containing 1 mg/mL LPS/LTA to final concentrations of $0.5\times$ MIC, $1\times$ MIC, $2\times$ MIC, $4\times$ MIC, and $8\times$ MIC, and the cultures were incubated for 2 h. Then 10-fold serial dilutions were prepared, and CFUs were counted. The experiment was repeated independently three times.

**Endotoxin neutralization experiments.** A standard curve was established according to the instructions provided with the kit (Genscript, China). A total of 100 $\mu$L of peptides at various concentrations was added to endotoxin-free tubes, followed by the addition of 100 $\mu$L of horseshoe crab reagent, and the mixture was incubated for 30 min at 37°C with protection from light. Then, 100 $\mu$L of color development solution was added, the tubes were incubated for an additional 15 min, and the absorbance at 545 nm was measured.

**Detection of inflammation *in vivo* and *in vitro*.** HeLa cells ($2.5 \times 10^5$) were inoculated into 24-well plates and incubated until the cells covered the bottom of the plate. LTA (200 ng/mL) and various concentrations of Tsap peptide were then added, and the cultures were incubated overnight in a $CO_2$ incubator at 37°C. The cell culture supernatant was then removed, centrifuged at 1,000 $\times$ $g$ for 10 min at 4°C, and TNF-$\alpha$ and IL-6 concentrations were measured by human TNF-$\alpha$ and IL-6 enzyme-linked immunosorbent assay (ELISA) kits (Ruixin Biotech, China).

**Animal experiments.** The animal experiments described in this paper were conducted in accordance with the standards and specifications for animal experiments of Huazhong Agricultural University, with ethical approval number HAZUMO-2020-0013. All procedures complied with animal welfare and protection regulations.

The LTA-induced mice model was used to detect inflammation *in vivo*. Fifteen BALB/c female mice weighing 18 $\pm$ 2 g each were divided randomly into three groups. The mice in one of the groups were given 20 mg/kg of body weight of LTA by intranasal administration, and those in the second group were given 20 mg/kg LTA and 20 mg/kg Tsap-1 by intranasal administration. The mice in the last group were given equal amounts of PBS administered in the same way. The lung tissue of the animals was then fixed in an electron microscope fixative and sectioned for microscopic observation, and images were obtained. All the pathological images were obtained through the same microscope (Olympus, Japan), and any potential observed lines are likely the result of an artifact in the Illustrator software (Adobe, USA) or microscope.

To determine the drug protection rate fraction, *S. aureus* USA200 that had been cultured overnight was cultured in LB broth to an $OD_{600}$ of 0.5, washed three times, and resuspended in PBS. Forty female BALB/c mice (18 $\pm$ 2 g each) were divided randomly into four groups. All four groups ($n = 10$) of mice were injected intraperitoneally with $1 \times 10^7$ bacteria. Six hours after infection, the animals in the four groups received 20 mg/kg Tsap-1, 20 mg/kg ciprofloxacin, 10 mg/kg Tsap-1, and 10 mg/kg ciprofloxacin or an equal amount of PBS by intraperitoneal injection. The survival rate of the mice was recorded after 7 days.

For the mouse tissue load experiment, the bacteria were manipulated as described previously. Twenty female BALB/c mice (18 $\pm$ 2 g each) were divided randomly into four groups of five mice each. The animals in the four groups were injected intraperitoneally with 20 mg/kg Tsap-1 AMP, 20 mg/kg ciprofloxacin, 10 mg/kg Tsap-1, and 10 mg/kg ciprofloxacin or an equal amount of PBS. Twelve hours later, the mice were anesthetized with ether and perfused with PBS, and their livers, spleens, lungs, and kidneys were harvested. The tissues were then homogenized using a tissue grinder. Then, $10\times$ serial dilutions of the homogenates were prepared and plated onto LB plates, and the CFUs were counted.

**Statistical analysis.** The above data were analyzed using GraphPad Prism 7.0. Student's *t* test was used to compare the means between two specific groups, and analysis of variance (ANOVA) was applied to compare the means of more than two groups (*, $P < 0.05$; **, $P < 0.01$; ***, $P < 0.001$; ns, not significant).

**Data availability.** The data presented in this study are available on request from the corresponding author.

## SUPPLEMENTAL MATERIAL

Supplemental material is available online only.
**SUPPLEMENTAL FILE 1**, DOC file, 0.04 MB.

## ACKNOWLEDGMENTS

Methodology, W.L., H.L., C.W., and C.T.; validation, W.L., H.L., C.T., G.W., C.W., and W.D.; formal analysis, W.L., H.L., C.T., writing-original draft preparation, W.L.; writing-review and editing, W.L. and C.T.; funding, C.T., All authors have read and agreed to the published version of the manuscript.

This work was funded by the National Key Research and Development Program (2021YFD1800402 and 2022YFD1800901), the earmarked fund for CARS-35, The Hubei Provincial Science and Technology Major Project (2022ABA002), and by Walmart Food Safety Collaboration Center of Walmart Foundation (project number 61626817).

We declare no conflict of interest.

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
