## [Reviewer comments · Microbiology Spectrum]

Microbiology Spectrum

Effectors of the type VI secretion system have the potential to be modified into antimicrobial peptides

Wenjia Lu, hao Lu, Chenchen Wang, Gaoyan Wang, wen dong, and Chen Tan

Corresponding Author(s): Chen Tan, Huazhong Agricultural University

Review Timeline:

Submission Date:	January 19, 2023
Editorial Decision:	March 21, 2023
Revision Received:	April 7, 2023
Accepted:	April 26, 2023

Editor: Cesar de la Fuente-Nunez

Reviewer(s): Disclosure of reviewer identity is with reference to reviewer comments included in decision letter(s). The following individuals involved in review of your submission have agreed to reveal their identity: Renee N Fleeman (Reviewer #3)

Transaction Report:

DOI: <https://doi.org/10.1128/spectrum.00308-23>

March 21, 2023

Dr. Chen Tan
Huazhong Agricultural University
Wuhan
China

Re: Spectrum00308-23 (Effectors of the type VI secretion system have the potential to be modified into antimicrobial peptides)

Dear Dr. Chen Tan:

Link Not Available

Sincerely,

Cesar de la Fuente

Journals Department
Reviewer comments:

Reviewer #3 (Comments for the Author):

The authors present a detailed analysis of the potential for type 6 secretion system effectors as antimicrobials. This is a thorough investigation using co-culturing assays to identify the effectors with potential, followed by a robust study of mechanisms and structural characteristics. They use multiple measures to show membrane disruption and were able to link the increased gram-positive antimicrobial activity to the strong interaction with LTA compared to LPS. They have done a large amount of research into these peptides and effectively show that type 6 secretion system effectors have potential as antimicrobials. Therefore, I have only one major concern and a few minor comments about the antimicrobial assay methods. There is also a need for more explanation in the text for certain results. I have listed these concerns below:

Major comment:

1. The minimum bactericidal activity of the peptides was not done in a way that can assess bactericidal activity. The MBC is quantified from bacterial recovery following removal of the therapeutic. The authors explain the MBC as the concentration in agar that inhibits bacterial growth. This does not allow one to see if the bacteria can recover from the drug effect following its removal.

In addition, there are some concerns with the antimicrobial assays to determine MICs. The 12-hour growth time for these antimicrobial assays is shorter than the standard 18-20 hours. And the positive controls explained in the methods show 200 μ L of 1×10^6 bacteria where all drug treatment will have half this amount because of the addition of medium containing the drugs (dilutes bacteria in half). This gives you more bacteria to start with in the positive control wells.

Minor comment:

1. ExPEC RS218 is never spelled out as Extraintestinal pathogenic *Escherichia coli*. This should be done before using the abbreviation throughout. Especially in the supplemental table, *E. coli* is not listed for many strains that are such. And *S. aureus* USA200 is used sometimes while other times it is called *S. aureus* 200. There are also instances within the text where a strain name is mentioned, and it is not clear that it is *E. coli*.
2. Similar to comment 1: Tsap is first mentioned in the introduction but not until the end of the first section of the results is it explained that this is what Tsap stands for. It should be in the first mention of this acronym that it is explained.
3. The authors change between G+ in some sections and gram-positive in others. It is best to be consistent with these nomenclatures throughout the text.
4. LA is used in the text without explanation of what it is. I believe this may be LBA being luria-bertani agar because LB broth is used. If so, the common abbreviation for this is LBA not LA.
5. Line 106 MICs is not defined as it is the first use of this abbreviation within the text
6. There is very little discussion of the MIC and MBC results. One sentence for the results of the MDR testing presented in table 2 does not give the reader an details into the results found.
7. Line 115 says "Antipeptide" I believe this is supposed to be antimicrobial peptide?
8. Line 126, the figure legend calls them "Spiral" wheels but in the text they are called "helical" Consistency will help the reader understanding.
9. Lines 149-152 describes the SIM analysis. There is very minimal explanation of the results and no figure mentioned here. There also should be an explanation of what SIM is and how it is better for the analysis than regular confocal or fluorescent microscopy.
10. Lines 157-161 describe the hydrogen ion content and ROS. There is not enough detail about what these assays are measuring and why it is important for you to measure this. (in contrast there is a great explanation of the assay for DIOC2(3) fluorescent probe)
11. Line 176 says transmission electron microscopy was used but the figure and methods shows scanning microscopy (it is clearly scanning from the figure). Also, figure 6H should be mentioned in the first sentence describing the results on line 178.
12. Line 200 ITC is not defined
13. Lines 204-209 do not reference the specific figures panels corresponding to the data being discussed.
14. Line 222 says "LPS was almost completely neutralized" This is the only mention of LPS in this section about LTA. Is this supposed to be LTA?
15. In lines 228-233 there is no mention in the text of the figures corresponding to the endotoxemia model.
16. Lines 258 and 260 use the term "orders of magnitude". This is not the normal scientific language used to describe changes in data and is therefore confusing. It would be better to present this data as fold-change which is what researchers generally use.
17. Line 272 *Bacillus subtilis* is a different size font that the other words.
18. Line 305, HBMECs are not introduced as to what this stands for.
19. Line 337-341, the asterisks should be replaced with x as is commonly used for these purposes within text.
20. The SEM methods does not say the brand or type of SEM that was used.
21. The endotoxin neutralization experiments should say exactly which kit was used and list the manufacturer.
22. Lines 449-450 do not describe how the supernatant was assayed for TNF-a and IL-6. There needs to be an explanation of this experiment.
23. Table 1 and 3: rows and columns labels are a bit messy with words and strain names cut in half.
24. Meletin is used as a control for the toxicity studies but is not mentioned in the results text and figure legend. All data presented should be mentioned in the text, especially controls.
25. The words on Figure 1 are pixelated compared to all other figures. Please be sure all figures are good resolution.

Staff Comments:

Preparing Revision Guidelines

To submit your modified manuscript, log onto the eJP submission site at <https://spectrum.msubmit.net/cgi-bin/main.plex>. Go to Author Tasks and click the appropriate manuscript title to begin the revision process. The information that you entered when you

first submitted the paper will be displayed. Please update the information as necessary. Here are a few examples of required updates that authors must address:

Please return the manuscript within 60 days; if you cannot complete the modification within this time period, please contact me. If you do not wish to modify the manuscript and prefer to submit it to another journal, please notify me of your decision immediately so that the manuscript may be formally withdrawn from consideration by Microbiology Spectrum.

Dear editor,

I delayed my lumbar hernia treatment for a week because of the response.

I am sorry.

My opinion for publication is to be accepted. The study is very up-to-date, accurate and of a quality that will shed light on future studies. Therefore, the publication has been accepted by me.

Associate Professor Demet Celebi

Dear editor,

I delayed my lumbar hernia treatment for a week because of the response.

I am sorry.

My opinion for publication is to be accepted. The study is very up-to-date, accurate and of a quality that will shed light on future studies. Therefore, the publication has been accepted by me.

Associate Professor Demet Celebi

Effectors of the type VI secretion system have the potential to be modified into antimicrobial peptides

The authors present a detailed analysis of the potential for type 6 secretion system effectors as antimicrobials. This is a thorough investigation using co-culturing assays to identify the effectors with potential, followed by a robust study of mechanisms and structural characteristics. They use multiple measures to show membrane disruption and were able to link the increased gram-positive antimicrobial activity to the strong interaction with LTA compared to LPS. They have done a large amount of research into these peptides and effectively show that type 6 secretion system effectors have potential as antimicrobials. Therefore, I have only one major concern and a few minor comments about the antimicrobial assay methods. There is also a need for more explanation in the text for certain results. I have listed these concerns below:

Major comment:

1. The minimum bactericidal activity of the peptides was not done in a way that can assess bactericidal activity. The MBC is quantified from bacterial recovery following removal of the therapeutic. The authors explain the MBC as the concentration in agar that inhibits bacterial growth. This does not allow one to see if the bacteria can recover from the drug effect following its removal.

In addition, there are some concerns with the antimicrobial assays to determine MICs. The 12-hour growth time for these antimicrobial assays is shorter than the standard 18-20 hours. And the positive controls explained in the methods show 200 μ L of 1×10^6 bacteria where all drug treatment will have half this amount because of the addition of medium containing the drugs (dilutes bacteria in half). This gives you more bacteria to start with in the positive control wells.

Minor comment:

1. ExPEC RS218 is never spelled out as Extraintestinal pathogenic *Escherichia coli*. This should be done before using the abbreviation throughout. Especially in the supplemental table, *E. coli* is not listed for many strains that are such. And *S. aureus* USA200 is used sometimes while other times it is called *S. aureus* 200. There are also instances within the text where a strain name is mentioned, and it is not clear that it is *E. coli*.
2. Similar to comment 1: Tsap is first mentioned in the introduction but not until the end of the first section of the results is it explained that this is what Tsap stands for. It should be in the first mention of this acronym that it is explained.
3. The authors change between G+ in some sections and gram-positive in others. It is best to be consistent with these nomenclatures throughout the text.
4. LA is used in the text without explanation of what it is. I believe this may be LBA being luria-bertani agar because LB broth is used. If so, the common abbreviation for this is LBA not LA.

5. Line 106 MICs is not defined as it is the first use of this abbreviation within the text
6. There is very little discussion of the MIC and MBC results. One sentence for the results of the MDR testing presented in table 2 does not give the reader an details into the results found.
7. Line 115 says "Antipeptide" I believe this is supposed to be antimicrobial peptide?
8. Line 126, the figure legend calls them "Spiral" wheels but in the text they are called "helical" Consistency will help the reader understanding.
9. Lines 149-152 describes the SIM analysis. There is very minimal explanation of the results and no figure mentioned here. There also should be an explanation of what SIM is and how it is better for the analysis than regular confocal or fluorescent microscopy.
10. Lines 157-161 describe the hydrogen ion content and ROS. There is not enough detail about what these assays are measuring and why it is important for you to measure this. (in contrast there is a great explanation of the assay for DIOC2(3) fluorescent probe)
11. Line 176 says transmission electron microscopy was used but the figure and methods shows scanning microscopy (it is clearly scanning from the figure). Also, figure 6H should be mentioned in the first sentence describing the results on line 178.
12. Line 200 ITC is not defined
13. Lines 204-209 do not reference the specific figures panels corresponding to the data being discussed.
14. Line 222 says "LPS was almost completely neutralized" This is the only mention of LPS in this section about LTA. Is this supposed to be LTA?
15. In lines 228-233 there is no mention in the text of the figures corresponding to the endotoxemia model.
16. Lines 258 and 260 use the term "orders of magnitude". This is not the normal scientific language used to describe changes in data and is therefore confusing. It would be better to present this data as fold-change which is what researchers generally use.
17. Line 272 *Bacillus subtilis* is a different size font that the other words.
18. Line 305, HBMECs are not introduced as to what this stands for.
19. Line 337-341, the asterisks should be replaced with x as is commonly used for these purposes within text.
20. The SEM methods does not say the brand or type of SEM that was used.
21. The endotoxin neutralization experiments should say exactly which kit was used and list the manufacturer.
22. Lines 449-450 do not describe how the supernatant was assayed for TNF-a and IL-6. There needs to be an explanation of this experiment.
23. Table 1 and 3: rows and columns labels are a bit messy with words and strain names cut in half.
24. Meletin is used as a control for the toxicity studies but is not mentioned in the results text and figure legend. All data presented should be mentioned in the text, especially controls.
25. The words on Figure 1 are pixelated compared to all other figures. Please be sure all figures are good resolution.

Response to Reviewers

Dear Editors and Reviewers:

Thank you for your letter and for the reviewers' comments concerning our manuscript entitled "Effectors of the type VI secretion system have the potential to be modified into antimicrobial peptides". Those comments are all valuable and very helpful for revising and improving our paper, as well as the important guiding significance to our research. We have studied the comments carefully and have made corrections which we hope to meet with approval. Revised portions are highlighted in yellow in the paper. The main corrections in the paper and the responds to the reviewer's comments are as flowing: Responds to the reviewer's comments:

Reviewer comments:

Reviewer #3 (Comments for the Author):

The authors present a detailed analysis of the potential for type 6 secretion system effectors as antimicrobials. This is a thorough investigation using co-culturing assays to identify the effectors with potential, followed by a robust study of mechanisms and structural characteristics. They use multiple measures to show membrane disruption and were able to link the increased gram-positive antimicrobial activity to the strong interaction with LTA compared to LPS. They have done a large amount of research into these peptides and effectively show that type 6 secretion system effectors have potential as antimicrobials. Therefore, I have only one major concern and a few minor comments about the antimicrobial assay methods. There is also a need for more explanation in the text for certain results. I have listed these concerns below:

Dear reviewer, thank you for reviewing our manuscript and for the constructive comments, which greatly helped us to improve the manuscript. We have heavily revised our experiments. The manuscript was carefully revised and point-by-point response was listed below. We hope that your comments have been addressed

accurately. The revised manuscript was highlighted with yellow color and the responses were presented in blue text. The page size was changed in accordance with the ASM journal's editorial style (<https://journals.asm.org/editorial-style>), hence the line numbers may have changed somewhat. Nonetheless, the new version has the proper line numbers.

Major comment:

1. The minimum bactericidal activity of the peptides was not done in a way that can assess bactericidal activity. The MBC is quantified from bacterial recovery following removal of the therapeutic. The authors explain the MBC as the concentration in agar that inhibits bacterial growth. This does not allow one to see if the bacteria can recover from the drug effect following its removal.

Answer: It is true as Reviewer suggested, our protocol is tested as the manuscript “Xiao X, Lu H, Zhu W, Zhang Y, Huo X, Yang C, Xiao S, Zhang Y, Su J. A Novel Antimicrobial Peptide Derived from Bony Fish IFN1 Exerts Potent Antimicrobial and Anti-Inflammatory Activity in Mammals. *Microbiol Spectr.* 2022 Apr 27;10(2):e0201321. doi: 10.1128/spectrum.02013-21. Epub 2022 Mar 15. PMID: 35289673; PMCID: PMC9045357.” described before. But just like the review pointed out, we already changed the protocol in lines 290-291 and renewed the result in Table 1-2.

In addition, there are some concerns with the antimicrobial assays to determine MICs. The 12-hour growth time for these antimicrobial assays is shorter than the standard 18-20 hours. And the positive controls explained in the methods show 200 μ L of 1×10^6 bacteria where all drug treatment will have half this amount because of the addition of medium containing the drugs (dilutes bacteria in half). This gives you more bacteria to start with in the positive control wells.

Answer: Thanks a lot for your constructive advice, we retested the MIC according to the standard protocol, the result is the same as our result presented before. The mistake of positive control is because of our incorrect description, we already changed it in lines 283-284.

Minor comment:

1. ExPEC RS218 is never spelled out as Extraintestinal pathogenic *Escherichia coli*. This should be done before using the abbreviation throughout. Especially in the supplemental table, *E. coli* is not listed for many strains that are such. And *S. aureus* USA200 is used sometimes while other times it is called *S. aureus* 200. There are also instances within the text where a strain name is mentioned, and it is not clear that it is *E. coli*.

Answer: We are very sorry for our negligence of this mistake, we already spelled out *Extraintestinal pathogenic Escherichia coli (ExPEC)* in line 40 and renewed the information of strains both in each strain of Table S1 and in the manuscript. *S. aureus* 200 is *S. aureus* USA200, we already changed in the text and Table S1. We also revised *S. aureus* 300 to *S. aureus* USA300.

2. Similar to comment 1: Tsap is first mentioned in the introduction but not until the end of the first section of the results is it explained that this is what Tsap stands for. It should be in the first mention of this acronym that it is explained.

Answer: Thanks a lot for your kind remind, and we already changed this mistake in line 56 “Type VI secretion system-related antibacterial peptide (Tsap)”.

3. The authors change between G⁺ in some sections and gram-positive in others. It is best to be consistent with these nomenclatures throughout the text.

Answer: We are very sorry for our incorrect writing, we changed G⁺ to gram-positive in lines 59, 451, 454 and 477. Also changed G⁻ to gram-negative in lines 451 and 477.

4. LA is used in the text without explanation of what it is. I believe this may be LBA being luria-bertani agar because LB broth is used. If so, the common abbreviation for this is LBA not LA.

Answer: We are very sorry for our incorrect writing, we already revised LA to LBA in lines 226, 248 and 632.

5. Line 106 MICs is not defined as it is the first use of this abbreviation within the text
Answer: We are very sorry for our negligence of this mistake, we added “minimal inhibitory concentration” in line 88.

6. There is very little discussion of the MIC and MBC results. One sentence for the

results of the MDR testing presented in table 2 does not give the reader an details into the results found.

Answer: We are very sorry for our negligence of this part, we already added the result in lines 95-102.

7. Line 115 says "Antipeptide" I believe this is supposed to be antimicrobial peptide?

Answer: We are very sorry for our incorrect writing, we already revised this misspelling in line 95 and line 130.

8. Line 126, the figure legend calls them "Spiral" wheels but in the text they are called "helical" Consistency will help the reader understanding.

Answer: We are very sorry for our negligence of this mistake, we changed "helical" to "Spiral" in line 107.

9. Lines 149-152 describes the SIM analysis. There is very minimal explanation of the results and no figure mentioned here. There also should be an explanation of what SIM is and how it is better for the analysis than regular confocal or fluorescent microscopy.

Answer: It is really true as Reviewer suggested that this part need more explanation, we already added explanation of the results and explanation of what SIM is in line 129-134.

10. Lines 157-161 describe the hydrogen ion content and ROS. There is not enough detail about what these assays are measuring and why it is important for you to measure this. (in contrast there is a great explanation of the assay for DIOC2(3) fluorescent probe)

Answer: Thanks a lot for your constructive advice, we added the explanation of why we detect the content of hydrogen ion in lines 136-139 and explained the ROS detect reason in lines 141-143.

11. Line 176 says transmission electron microscopy was used but the figure and methods shows scanning microscopy (it is clearly scanning from the figure). Also, figure 6H should be mentioned in the first sentence describing the results on line 178.

Answer: We are very sorry for our negligence in this mistake, we already corrected it in lines 160, 665, and 666. We already mentioned Figure 6H in line 160.

12. Line 200 ITC is not defined

Answer: We are very sorry for our negligence in this mistake, we already defined “isothermal titration calorimetry (ITC)” in line 168.

13. Lines 204-209 do not reference the specific figures panels corresponding to the data being discussed.

Answer: We are very sorry for our negligence of this mistake, we already added in line 174.

14. Line 222 says "LPS was almost completely neutralized" This is the only mention of LPS in this section about LTA. Is this supposed to be LTA?

Answer: We are very sorry for our negligence of this mistake, we changed “LPS” to “endotoxin” in line 182.

15. In lines 228-233 there is no mention in the text of the figures corresponding to the endotoxemia model.

Answer: We are very sorry for our negligence of this mistake, we already added in lines 183-184 and line 189.

16. Lines 258 and 260 use the term "orders of magnitude". This is not the normal scientific language used to describe changes in data and is therefore confusing. It would be better to present this data as fold-change which is what researchers generally use.

Answer: We are very sorry for our miswriting, and we already changed in line 212 “reduced by about 32 times reduced by about 32 times” and line 213 “decreased by 1000-fold”.

17. Line 272 Bacillus subtilis is a different size font than the other words.

Answer: We are very sorry for our negligence of this mistake, we already corrected this mistake in line 219.

18. Line 305, HBMECs are not introduced as to what this stands for.

Answer: thanks for your kind remind, HBMECs means “Human Brain Microvascular Endothelial Cells”, we already added it in line 253.

19. Line 337-341, the asterisks should be replaced with x as is commonly used for these purposes within text.

Answer: We are very sorry for our negligence of this mistake, we already corrected this mistake in our manuscript and highlighted it in lines 257, 269, 274, 283, 284, 287, 295, 297, 312, 320, 362, 379, 380, 391, 394, 412, 423, 623, 626, 632, 645, 651, 661, 664, and 676.

20. The SEM methods does not say the brand or type of SEM that was used.

Answer: We are very sorry for our negligence of this mistake, we already added the type of SEM we used “HITACHI Regulus 8100, Japan” in line 365.

21. The endotoxin neutralization experiments should say exactly which kit was used and list the manufacturer.

Answer: We are very sorry for our negligence of this mistake, we already listed Item No. and manufacturer “Genscript, China” in line 385.

22. Lines 449-450 do not describe how the supernatant was assayed for TNF- α and IL-6. There needs to be an explanation of this experiment.

Answer: We are very sorry for our negligence in this mistake, we already added “TNF- α and IL-6 concentrations measured by Human TNF- α and IL-6 ELISA kits (RUIXIN BIOTECH, China).” In lines 395-396.

23. Table 1 and 3: rows and columns labels are a bit messy with words and strain names cut in half.

Answer: We are very sorry for our negligence of this mistake, we already adjusted the distance of each row in Table 1 and Table 3.

24. Meletin is used as a control for the toxicity studies but is not mentioned in the results text and figure legend. All data presented should be mentioned in the text, especially controls.

Answer: We are very sorry for our negligence of this part, and we already added “compared with positive control melittin treated group”, “But 4 ug/mL melittin treated group already showed complete hemolysis activity”, “0 ug/mL to 512 ug/mL of melittin were used as positive control, and PBS group as negative control.” and “and melittin was used as the control” in the manuscript and figure legend in line 646-649.

25. The words on Figure 1 are pixelated compared to all other figures. Please be sure

all figures are good resolution.

Answer: It is true as Reviewer suggested that the words in Figure 1 are pixelated compared to all other figures. This is because in our initial submission, more than 100MB files cannot be uploaded, so we decreased the resolution of the figures, and we already uploaded the original figures this time.

We tried our best to improve the manuscript and made some changes in the manuscript. These changes will not influence the content and framework of the paper. We appreciate for Editors/Reviewers' warm work earnestly and hope that the correction will meet with approval.

Once again, thank you very much for your comments and suggestions.

April 26, 2023

Dr. Chen Tan
Huazhong Agricultural University
Wuhan
China

Re: Spectrum00308-23R1 (Effectors of the type VI secretion system have the potential to be modified into antimicrobial peptides)

Dear Dr. Chen Tan:

Your manuscript has been accepted, and I am forwarding it to the ASM Journals Department for publication. You will be notified when your proofs are ready to be viewed.

Sincerely,

Cesar de la Fuente-Nunez
Editor, Microbiology Spectrum
